# Valorization of Bioactive Compounds from By-Products of *Matricaria recutita* White Ray Florets

**DOI:** 10.3390/plants12020396

**Published:** 2023-01-14

**Authors:** Ilva Nakurte, Marta Berga, Laura Pastare, Liene Kienkas, Maris Senkovs, Martins Boroduskis, Anna Ramata-Stunda

**Affiliations:** 1Institute for Environmental Solutions, “Lidlauks”, Priekuli Parish, LV-4126 Cesis, Latvia; 2Field and Forest, SIA, 2 Izstades Str., Priekuli Parish, LV-4126 Cesis, Latvia; 3Microbial Strain Collection of Latvia, Faculty of Biology, University of Latvia, 1 Jelgavas Str., LV-1004 Riga, Latvia; 4Alternative Plants, SIA, 2 Podraga Str., LV-1023 Riga, Latvia

**Keywords:** *Matricaria recutita*, white ray florets, CO_2_ extraction, valorization, chemical composition, cytotoxicity, phototoxicity, antimicrobial activity

## Abstract

In this research, we have reported the valorization possibilities of *Matricaria recutita* white ray florets using supercritical fluid extraction (SFE) with CO_2_. Experiments were conducted at temperatures of 35–55 °C and separation pressures of 5–9 MPa to evaluate their impact on the chemical composition and biological activity of the extracts. The total obtained extraction yields varied from 9.76 to 18.21 g 100 g^−1^ DW input. The greatest extraction yield obtained was at 9 MPa separation pressure and 55 °C in the separation tank. In all obtained extracts, the contents of total phenols, flavonoids, tannins, and sugars were determined. The influence of the supercritical CO_2_ extraction conditions on the extract antioxidant capacity was evaluated using the quenching activity of 2,2-diphenyl-1-picrylhydrazyl (DPPH). The chemical composition of the extracts was identified using both gas and liquid chromatography–mass spectrometry methods, whereas analyses of major and minor elements as well as heavy metals by microwave plasma atomic emission spectrometer were provided. Moreover, extracts were compared with respect to their antimicrobial activity, as well as the cytotoxicity and phototoxicity of the extracts. The results revealed a considerable diversity in the phytochemical classes among all extracts investigated in the present study and showed that the *Matricaria recutita* white ray floret by-product possesses cytotoxic and proliferation-reducing activity in immortalized cell lines, as well as antimicrobial activity. To the best of our knowledge, this is the first paper presenting such comprehensive data on the chemical profile, antioxidant properties, and biological properties of SFE derived from *Matricaria recutita* white ray florets. For the first time, these effects have been studied in processing by-products, and the results generated in this study provide valuable preconditions for further studies in specific test systems to fully elucidate the mechanisms of action and potential applications, such as potential use in cosmetic formulations.

## 1. Introduction

Making industrial processes and products with the least amount of environmental impact is idea behind green chemistry. Anastas and Warner’s twelve principles of green chemistry are essentially the sustainability principles for chemistry and related businesses [1]. Management of herbal waste byproducts resulting from various medicinal plant industrial cycles is poorly defined. Generated byproducts and waste have a detrimental influence on the environment. Many of these streams are underutilized and end up in landfills, where they contribute to greenhouse gas emissions, can cause further problems owing to microbial decomposition and leachate formation, and contribute to greenhouse gas emissions [2]. Biomass is sometimes used in enterprises to create energy or to create compost [2,3]. Instead of creating new revenue streams, more expenditures are incurred to manage solid waste in landfills because resources are handled as waste from an economic standpoint. In addition, the handling of vast quantities of various degradable materials is a challenge [4]. Due to the antigerminative properties of some fragrant plants, which may also be present in the plant waste, recycling the byproduct to energy requires a significant financial investment, and recycling to composting is not always appropriate [5]. It is generally recognized that the plant’s aerial parts have the capacity to serve as natural antioxidants, scavenging free radicals with their rich polyphenol and terpene content. This suggests that byproducts from the industrial processing of medicinal plants may include bioactive compounds that may be used for a variety of applications. As the need for extractable compounds develops, so does interest in recovering valuable compounds from various industrial waste streams [6].

Innovative, health-promoting, plant-based additives and ingredients have become more popular in the cosmetics business over the past several years. Polyphenols and other bioactive chemicals are of interest to the cosmetic industry because they may be used to improve different aspects of skin health. This hugely diverse group of chemicals also possesses numerous biological activities, including UV protection, antioxidant and anti-inflammatory activity, regulation of the skin microbiome, and synthesis of the skin extracellular matrix [2,7,8]. Present findings indicate that residual sage *(Salvia officinalis*) contains considerable levels of flavonoids responsible for antioxidative, antibacterial, and antifungal activities, such as apigenin and luteolin [9]. Waste of lavandin (*Lavandula x intermedia*) and spike lavender (*Lavandula latifolia*) are rich sources of phenolic compounds, such as phenolic acids and flavonoids, many of which are strong antioxidants and antimicrobials [10]. Extracts of residual rosemary (*Rosmarinus officinalis*), savory, and oregano species are also a good source of natural antioxidants, especially phenolic acids such as caffeic and rosmarinic acids, and a phenolic diterpene called carnosic acid [11]. From the residual clary sage (*Salvia sclarea*), a diterpene sclareol can be extracted, which is an important precursor for the synthesis of ambrox, the crucial aromatic compound in ambergris and a staple in the contemporary perfume industry. Waste obtained after steam distillation of chamomile (*Matricaria recutita*) could be a source of water-soluble pectic polysaccharides [12], as well as processing waste that remains after chamomile processing in significant amounts and is considered a rich source of apigenin and ferulic acid derivatives, as well as coumarin derivatives, herniarin and umbelliferone. These compounds are often used in the cosmetic industry due to their strong absorption of UV light [13].

Knowledge of phytochemical constituents and their biological activities in by-products is valuable for authentication and is the first step towards studying the biological activities of the whole product. By-product extracts can contain various types of bioactive compounds with different polarities, which makes it hard to identify and characterize.

Bioactive compounds present in the medicinal plant by-products can be separated individually from the biological matrix using different physical or chemical extraction techniques. The need for more efficient and environmentally friendly extraction encourages the development of new methods and technologies. Extraction processes are affected by several factors, including the technique used, the raw material, and the organic solvent.

Conventional techniques such as maceration, percolation, and Soxhlet extraction generally require large amounts of organic solvents, a lot of energy, and a lot of time, which has sparked interest in new technologies known as clean or green technologies. Supercritical fluid extraction, pressurized liquid extraction, ultrasound- and microwave-assisted extraction, and pulse electric field extraction are examples of these technologies. These techniques have the potential to reduce or eliminate the use of toxic solvents, thereby protecting the natural environment and its resources [14]. Supercritical fluid extraction (SFE) with CO_2_ is considered a sustainable method for the extraction of plant materials as carbon dioxide is non-toxic, harmless to humans and the environment, and it can be recycled during the extraction process. SFE is mainly used in the food industry for the decaffeination of coffee and the deodorization of fish oils, and in the pharmaceutical industry for the extraction of bioactive compounds, but can also be used in other industries such as environmental engineering, which assists in soil remediation [15]. At a particular temperature and pressure point, gases become supercritical fluids; the gas and liquid phases are indistinguishable. Further changes in the supercritical fluid’s temperature and pressure alter its solvent properties for non-polar compound extraction. The SFE method is rapid, automatable, selective, and avoids the use of large amounts of toxic solvents. Due to the use of low temperature regimes, it can be considered suitable for thermosensitive compounds. The SFE method is incapable of extracting compounds with large molecular weights. The introduction of modifiers (co-solvents) into the system, such as ethanol, methanol, or water, enhances the solvating power of CO_2_. It may increase the selectivity and extraction yield of target compounds. The extraction process does not require further cleaning of the extract. No solvent residues are left in it as the CO_2_ evaporates completely. SFE equipment may also include several separation tanks with controlled temperature and pressure settings. Based on the reduction in pressure and change in temperature, different compounds will precipitate in each stage of separation. The separation may occur either because the supercritical fluid is no longer supercritical or because the solute is no longer soluble in the supercritical fluid. This step may concentrate and isolate compounds from the extract. The main advantage of the separation within the extraction process is time and resource savings. It can also be carried out on small amounts of biomass samples [16].

The aim of our study was to evaluate and compare the chemical composition, antimicrobial activity, cytotoxicity, and phototoxicity of supercritical fluid extracts of German chamomile processing waste. *Matricaria recutita* white ray florets were collected from a dry herb processing facility. Supercritical fluid extraction was carried out in a pilot-scale plant with carbon dioxide as a solvent and ethanol as a co-solvent. Phytochemical analyses of extracts were performed using high-throughput 96-well plate spectrophotometric methods, liquid chromatography–mass spectrometry, and gas chromatography–mass spectrometry methods. Microwave plasma atomic emission spectrometer analyses of major and minor elements, as well as heavy metals, were provided. Minimum inhibitory concentration (MIC) and minimum bactericidal concentration (MBC) were examined to find out the antimicrobial properties of the extracts. For extract safety profiling, standard cytotoxicity and phototoxicity assessments following OECD (The Organization for Economic Co-operation and Development) guidelines were performed. Statistical analysis was carried out in order to determine the significant correlations between the different extracts and to create models that can meaningfully predict the targeted phytochemical recovery and the most promising chemical and biological activities.

## 2. Results and Discussion

### 2.1. Supercritical Fluid Extraction

The determination of the extraction yield is an important parameter to estimate the efficiency of SFE in the recovery of bioactive compounds. The extraction yields from the white ray florets of German chamomile are shown in Table 1 and Figure 1.

The greatest extraction yield obtained was at 9 MPa separation pressure and 55 °C in the separation tank (sample FAF3). The highest overall extraction yields as obtained in the current research is higher than those previously reported. Extract yield between 0.23 and 3.64 g/100 g of different processed and unprocessed chamomile flowers has been reported by Molnar and team [13], whereas Kotnik [17] and Scalia [18] reported yields of 2.50–3.81 g/100 g and 9.2–9.7g/100 g, respectively, obtained from ground chamomile flower heads. The addition of ethanol as a co-solvent enhances the solvation power of supercritical carbon dioxide and improves the recovery of bioactive compounds, which will also appear in the following results.

### 2.2. Phytochemical Screening of Matricaria recutita White Ray Ffloret Supercritical Fluid Extracts

The solubility of obtained CO_2_ extracts in three different solvents at a concentration of 1:10 (*w*/*v*) was determined for testing purposes. It was observed that all CO_2_ extracts are partly soluble in water and ethanol, whereas they completely dissolve in cyclohexane. These observations are also confirmed by the gravimetrical analysis (Figure 2), which showed that the sum of essential oils (EO) and waxes in the tested CO_2_ extracts varied from 66.3% (FAF2), 76.72% (FAF1), and 78.17% (FAF3).

Based on the obtained data, it was decided that for phytochemical screening by high-throughput 96-well plate methods and LC-gTOF-MS, CO_2_ samples will be diluted in 70% ethanol (EtOH), whereas samples for testing by GC-MS should be diluted in cyclohexane.

Screening of phytochemical classes in CO_2_ ethanol extracts from chamomile herb processing by-products was performed using high-throughput 96-well plate spectrophotometric methods. Table 2 shows the data as the mean standard error of three independent experiments.

The results revealed a considerable diversity in the phytochemical classes among all three extracts investigated in the present study. Significant differences in total phenolic content (TPC) emerged among extracts, with the FAF2 extract showing the highest quantity (342.8 ± 17.3 mg of GAE/mL) and the FAF1 and FAF3 samples having the lowest content (98.1 ± 3.5 and 138.1 ± 3.7 mg of GAE/mL, respectively). Additionally, total flavonoid (TFC) content was rather uneven among all extracts, as it ranged from 45.6 ± 2.7 mg of APE/mL to 201.8 ± 11.3 mg of APE/mL. The extracts also differed in terms of total tannin content (TTC), although FAF1 and FAF3 contained almost the same content (46.4 ± 1.9 mg of TAE/mL and 49.8 ± 2.7 mg of TAE/mL, respectively) compared with FAF3 (128.8 ± 12.4 mg of TAE/mL). The highest amounts of TPC, TFC, and TTC were found in FAF2 extracts, accordingly showing the highest radical scavenging activity (66.5%). At the same time, the content of total sugars in the FAF2 extract in comparison with other extracts was the lowest (53.0 ± 2.2 mg of GLE/mL), suggesting that a concentration increase could negatively affect antiradical activity. Nevertheless, for a more correct interpretation of the antiradical activity, the groups of other active compounds shown in Figure 1 should also be considered. Among all studied samples, extract FAF2 contains not only the highest amount of non-volatile compounds (33.67%), but also the highest content of essential oils (17.70%). Many studies have shown a correlation between the phenolic content of plants and their antioxidant power, as well as the biological activities of essential oils [19,20,21]. A detailed discussion is presented in Section 2.7.

### 2.3. Qualitative Analysis of the Main Nonvolatile Plant Components in Supercritical Fluid Extracts of Matricaria recutita White Ray Florets

The untargeted screening and identification of nonvolatile phytocomponents from *Matricaria recutita* processing by-product CO_2_ extracts were conducted through LC-qTOF-MS in positive ESI+ mode. The obtained results show that the LC-qTOF-MS instrument is useful for determining and confirming the main extracted bioactive compounds. The most valuable aspect of the LC-qTOF-MS is its ability to provide accurate molecular mass information, allowing the empirical formulas to be matched against databases without the need for authentic standards. The tentative identification was based on high-resolution mass spectra, fragmentation, and databases such as Metlin and LipidMaps data. In total, ninety-six bioactive compounds were isolated from all three valorization extracts (Table 3). The results indicated that the major constituents of the investigated extracts are less polar prenol lipids, fatty acyls, lignans, and glycerophospahtes, followed by more polar flavonoids, phenolic acids, amino acids, and lactones. Significant amounts of *Matricaria recutita* waste are considered a rich source of apigenin and ferulic acid derivatives, as well as the coumarin derivatives herniarin and umbelliferone [13]. These compounds are often used in the cosmetics industry due to their strong absorption of UV light. Although the aim of this study was not to quantify the identified compounds, our observations confirm the significant presence of hydroxycoumarins such as umbelliferone, herniarin, and their derivatives, as well as a series of apigenin-, ferulic-acid-, and cinnamic-acid-derived flavonoids, for example.

### 2.4. Quantitative Analysis of the Main Volatile Plant Components in Supercritical Fluid Extracts of Matricaria recutita White Ray Florets

In total, forty-seven compounds were separated in the *Matricaria recutita* processing by-product CO_2_ cyclohexane extracts using gas chromatography–mass spectrometry. The main identified compounds depicted in Figure 3 and their relative amounts (%) are summarized in Table 4.

The chemical compositions of the CO_2_ extracts in cyclohexane were determined according to their retention times and spectrometric electronic library (NIST). The identity of the constituents was established using GC retention indices (RIs). Generally, the chemical composition of CO_2_ extracts showed the abundance of essential oil components (3.05 –17.7%), hydrocarbons (15.61–59.29%), pentacyclic triterpenes (11.35 –68.79%) and sterols (1.63–4.57%). The presence of essential oil compounds in CO_2_ extracts was already expected, whereas chamomile (*Matricaria recutita*) flowers themselves are a rich source of essential oils [22], even when obtained from dried white flower petals. The essential oil typically produced by steam distillation is characterized by its blue color because of the chamazulene present. Chamazulene is not present in the fresh flowers; therefore, in our study, the obtained CO_2_ extracts exhibited dark yellow (FAF1 and FAF3) and green (FAF2) hues (Table 1), which is consistent with prior findings [13,23,24]. Previous research on the essential oil amount in CO_2_ extracts showed that it varies between 0.5 and 4.5% [24]. In the framework of the same study, the author indicates the differences in both the amount and content of the essential oil at the different SFE times. Not only the yield but also the essential oil amount increases with pressure (100–120 bar), temperature (30 and 40 °C), and solvent flow rate.

According to the obtained data, sample FAF2 showed the biggest differences within the analysis of volatile compounds (Figure 4). Samples FAF1 and FAF3 showed a higher content of pentacyclic triterpenes, whereas FAF2 contained more hydrocarbons. Based on GC-MS data, prenol lipids (pentacyclic triterpenes) were not only separated but also identified, whereas using LC-qTOF-MS was more challenging. It was found that these extracts contain valuable amounts of lupeol, α-amyrin and β-amyrin. The versatility of the pentacyclic triterpenes provides a promising approach not only for the food and cosmetics industries but for diabetes management as well. Among the identified aliphatic hydrocarbons, some of them are described as exhibiting antibacterial activity [24]. Essential oils obtained by steam distillation from areal parts of *Matricaria recutita* play an important role in a wide range of biological activities that allow their use to prevent or treat diseases, as well as being extensively used in the food industry (e.g., as anti-microbial and flavoring agents in beverages), perfumery, and cosmetics (e.g., as anti-inflammatory and calming agents in skin care products) [19]. Not only chamazulene, but also β-Farnesene, Bisabolol Oxide A, α-Bisabolone Oxide A, and α-Bisabolone Oxide B, as well as the spiroethers cis-ene-yne-Dicycloether and Tonghaosu, have been reported to influence the antioxidant effects of EO [19,20,21].

### 2.5. Major and Minor Elements and Heavy Metal Screening in Supercritical Fluid Extracts of Matricaria recutita White Ray Florets

Elemental analysis of *Matricaria recutita* processing by-product white ray florets CO_2_ extracts appear to be an important part of the quality assurance system, assisting in determining the actual level of exposure to various elements. Essential and non-essential elements are well defined (e.g., Co, Cr, Fe, Mn, Mo, Ni, Se, Sn, V, and Zn are essential elements, whereas As, Cd, Pb, and Hg are unambiguously toxic elements) [25,26], and therefore it is important to have quality control of elemental analyses in by-product CO_2_ extracts in order to estimate concentration values that have no adverse influence on human health. The detection of major, minor elements, and heavy metals in *Matricaria recutita* processing by-product CO_2_ extracts was performed using a microwave plasma atomic emission spectrometer (MP-AES). The results are summarized in Table 5.

No significant differences were observed between all extracts tested. The elemental screening revealed the presence of iron, calcium, sodium, and, in smaller amounts, potassium, magnesium, manganese, zinc, and copper. The highest contents of essential heavy metals were established for iron (1170–1189.3 mg/kg). The analyzed samples contained no heavy metals or the elements molybdenum (Mo) or cobalt (Co). So far, no data have been found on the elements present in CO_2_ extracts obtained from dried white flower petals of chamomile, whereas investigations of the content of micro- and macro-elements in different chamomile flower products are described. High levels of B, Ca, Cu, Fe, and P were found in chamomile raw materials and infusions, detected by inductively coupled plasma optical emission spectrometry (ICP-OES) and atomic absorption spectrometry (AAS) [26]. While testing five different tea samples produced by different manufacturers and purchased at a local market in Serbia, it was observed that the most present macro-element was potassium, followed by sodium, calcium, and magnesium. From the group of trace elements, the presence of iron, copper, zinc, and manganese was detected in all samples [27]. The mineral element content of the chamomile flower heads grown in Austria and their recovery in differently prepared infusions presented elemental concentrations of potassium, calcium, and magnesium [28]. Only data obtained in a study by Mihaljev [29] using atomic absorption and mass spectrometry with inductively coupled plasma revealed higher levels of iron, manganese, and zinc in raw chamomile flowerheads. The data obtained in our studies confirms that micro- and macro-elements and their concentration levels in samples obtained using CO_2_ extraction differ from other extraction techniques. These findings confirm those of Pohl [30] that elements such as Al, Ca, Cu, Ba, Fe, Mn, Sr, Ti, and V, for which the extraction rates in water are relatively small, appear to be much more strongly bound or immobilized to the organic matrix of medicinal plants. Therefore, using stronger extraction methods can help extract elements with lower solubility.

### 2.6. Antimicrobial Activity Screening of Ethanol Extracts from Matricaria recutita White Ray Floret Supercritical Fluid Extracts

CO_2_ extracts from *Matricaria recutita* processing by-products inhibited Gram-positive bacteria such as *Staphylococcus aureus*, Gram-negative bacteria such as *Pseudomonas aeruginosa* and *Escherichia coli*, and yeast *Candida albicans*. All tested extracts showed antimicrobial activity; however, no significant differences in MIC and MBC concentrations were observed between the tested extracts (Table 6). *S. aureus* and *Candida albicans* were inhibited, but higher extract concentrations were required to inhibit Gram-negative bacteria. This is in line with previously published data on the different antibacterial activities of natural compounds against Gram-negative and Gram-positive bacteria. Overall, Gram-positive bacteria are more susceptible to plant-derived antimicrobial compounds. Gram-negative bacteria are considered to be less susceptible, as the outer membrane of the cell wall hinders penetration of various compounds [31,32]. For all three extracts, MICs against P.aeruginosa were lower compared with *E. coli*, indicating species-specific differences in susceptibility of Gram-negative bacteria.

The majority of compounds identified in the extracts have been reported in the literature to possess antimicrobial activity. Extracts FAF1 and FAF3 had the highest amyrin content, and amyrin-type triterpenoids have been shown to have inhibitory activity against bacteria and yeasts, including *S. aureus* and *C. albicans* [33]. Han and Lee [34] elucidated that amyrin’s antibacterial activity against *E. coli* is due to its ability to induce oxidative stress in bacteria. The strongest inhibitory and bactericidal activity of FAF1 against *E. coli* might be linked to the highest β-amyrin content among tested extracts. Another triterpenoid found in extracts, lupeol, is known to inhibit the growth of *S. aureus and C. albicans* [35]. It is hypothesized that it also contributes to the antifungal and antistaphylococcal activity of the FAF extracts. The effect of the total sugars in the extracts on antimicrobial, medicinal, and preservation properties also should be considered. Previous studies have also been conducted where concentrated sugar solutions have been shown to possess significant antimicrobial properties against pathogens such as *Staphylococcus aureus*, *Bacillus subtilis*, and *Pseudomonas aeruginosa* [36]. The results obtained by Mizzi [37] indicate that high sugar concentrations inhibit bacterial growth, whereas very low concentrations show the opposite effect, i.e., they stimulate bacterial growth, indicating that there is a threshold concentration upon which sugars cease to act as antimicrobial agents and become media instead.

### 2.7. Cytotoxicity and Phototoxicity Assessment of Matricaria recutita White Ray Floret Supercritical Fluid Extracts

It is important to show the safety profile of plant-derived extracts prior to testing for more specific biological activities. As a result, in vitro cytotoxicity and phototoxicity testing in the Balb/c 3T3 cell line were performed in accordance with OECD guidelines.

The cytotoxicity assay demonstrated that all three extracts have cell viability-decreasing activity. FAF1 extract reduced cell viability by more than 20% at concentrations exceeding 0.035 mg/mL. Concentrations above 0.14 mg/mL reduced cell viability by more than 90%. FAF2 was cytotoxic at concentrations above 0.062 mg/mL; at concentration levels of 0.24 mg/mL and higher, the reduction in cell viability exceeded 90%. Concentrations above 0.027 mg/mL were cytotoxic in case of the FAF3. The data are summarized in Figure 5A–C.

Negative effects on cell viability were also observed in the phototoxicity assay. Viability was rapidly reduced in cells incubated for 1h with extracts without UV irradiation (see Figure 5D–F, -UV samples). Interestingly, for FAF1 and FAF2, viability was slightly higher in UV-irradiated cell cultures compared with cells not exposed to UV. However, the differences were not statistically significant. An explanation for this might be the antioxidative activity of the extracts, which complement cellular antioxidative defense mechanisms induced by UV exposure.

Previously, the cytotoxic effects of various Chamomile extracts have been tested in different cell cultures. Sak [38] investigated the effects of Chamomile on melanocyte and oral epidermal cell lines, reporting a decrease in cell viability at concentrations less than 0.1 mg/mL. Other authors have used various cancerous cell lines for cell viability testing and observed toxic effects at low Chamomile extract concentrations [39]. The triterpenes lupeol and amyrins found in our *Matricaria recutita* processing by-product CO_2_ extracts are thought to contribute to the cytotoxic activity. Cytotoxicity and antiproliferative activity of lupeol have been shown in various cell cultures [40,41,42]. Similarly, Amyrin in vitro has demonstrated cytotoxicity and cell apoptosis-inducing activity [43,44].

Overall results show that *Matricaria recutita* processing by-product white ray floret CO_2_ extracts are rich in biologically active compounds that we have shown to have cytotoxic and proliferation-reducing activity in immortalized cell lines, as well as antimicrobial activity. For the first time, these effects have been studied in processing by-products, and the results generated in this study provide valuable preconditions for further studies in specific test systems to fully elucidate the mechanisms of action and potential applications.

### 2.8. Statistical Analysis

Statistical analysis using a heatmap dendrogram (Figure 6) and correlogram (Figure 7) was carried out to determine the significant correlations between the investigated extracts and to create models that can meaningfully predict the targeted phytochemical recovery and most promising chemical and biological activities. The heatmap in Figure 6 represents concentration levels of volatile compound content, average concentrations of total phenolic, total flavonoid, total tannin, and sugars, as well as cytotoxicity, phototoxicity, antiradical, and antibacterial activities among all *Matricaria recutita* processing by-product white ray floret CO_2_ extracts. The red color indicates higher concentrations and activities, whereas the orange-to-yellow color indicates lower corresponding values.

The data obtained in Figure 6 illustrate obvious differences among all three investigated supercritical extracts and confirms the hypothesis that even small changes in the *Matricaria recutita* processing by-product white ray florets supercritical extraction parameters (such as temperature and pressure) can affect not only the yield but also the chemical composition, which in turn affects the further properties of the extracts. The extract with the smallest total extraction yield (FAF2) showed the highest concentration of total phenolics (TPC), flavonoids (TFC), and tannins (TTC), which in turn reflected on the increased antiradical activity (ARA) of the extract. However, the values of other parameters such as cytotoxicity and phototoxicity were higher for FAF3 extracts, suggesting that the values of these parameters are influenced by some individual hydrocarbons (15–17) and triterpenes (20–23). According to antimicrobial activity, the best results were extrapolated from extract FAF1. The highest inhibitory and bactericidal activity of FAF1 against *E. coli* (MIC EC and MBC EC) could be attributed to the highest *β*-amyrin (25) content among tested extracts, as well as the presence of high total sugar concentrations.

Although with the help of a heatmap it is possible to visually evaluate the changes quite accurately, as the number of extracts increases, data interpretation becomes more complicated and unwanted inaccuracies can occur. When correlograms are used, the correlations between all variables are determined uniformly, regardless of the number of extracts used. Moreover, it can help choose targeted supercritical extraction conditions to obtain meaningful phytochemical recovery and the most promising chemical and biological activities.

To verify the strength of the association between several variables, a correlogram was constructed (Figure 7). Previous descriptions of strong correlations of cytotoxicity (IC 50) and phototoxicity (UV− and UV+) among some individual hydrocarbons (15–17) and lupeol and taraxerol type-triterpenes (20–23), as well as inhibitory effects against *S. aureus* (MIC SA) and *C. albicans* (MIC CA), were confirmed by the obtained correlogram. The matrix also allows for the assessment of the relationships between detected α- and β-amyrin (24 and 25) and inhibitory against *S. aureus* (MBC SA) and *P. aeruginosa* (MIC PA). Total sugars were highly positively correlated with inhibitory against *P. aeruginosa* (MBC PA) and *E. coli* (MIC EC and MBC EC), with a weaker influence on inhibitory against *S. aureus* (MBC SA) and *P. aeruginosa* (MIC PA). Matrix correlation supports the role of sterols (18 and 19) on inhibitory activity against yeasts, including *C. albicans* (MFC CA). Strong negative correlations between essential oil compounds (1–10) and almost all tested variables, except antiradical activity (ARA), were also confirmed.

## 3. Materials and Methods

### 3.1. Plant Materials

*Matricaria recutita* white ray florets (Figure 8) were collected from the SIA Field and Forest dried herb processing facility located in Priekuli, Latvia. SIA Field and Forest grows, harvests, dries, and processes organic *Matricaria recutita* chamomile. White ray florets are a waste stream originating from broken flowerheads during processing of the German chamomile herbal biomass. The industrial separation of white ray florets was carried out with a cyclone separator, ensuring great separation efficiency and the purity of the separated waste flow. The waste flow contains more than 95% of white ray florets and less than 5% of yellow disc florets. The moisture content of white ray florets was 11.20 ± 0.34% wt%, determined with a Radwag MA 210.X2.IC.A.WH moisture analyzer (Radom, Poland). Prior to the extraction, white ray florets were pulverized using a universal dry product mill, the MLVS M500 (Mazeikiai, Lithuania). The average particle diameter was 0.5 mm.

### 3.2. Supercritical Fluid Extraction

Supercritical fluid extraction was carried out with a pilot-scale supercritical fluid extractor, the CAREDDI SCFE-5L (UAB Analytical Solutions; Kaunas, Lithuania), using carbon dioxide (CO_2_) and ethanol (EtOH) as co-solvents. The supercritical extraction unit consisted of a high-pressure CO_2_ pump, a co-solvent pump, a cooling system, a heating system, a temperature control system, a CO_2_ storage tank, a 5 L extraction tank, and two-stage separation tanks (4 L and 2 L). The process diagram is depicted in Figure 9.

Flow rates and temperatures were automatically controlled. Manual back pressures were used to control the pressure in the extraction tank and both separation tanks. The maximum extraction pressure was 48 MPa, and the maximum separation pressure was 25 MPa. Dynamic extraction was used, with a CO_2_ flow rate of 100 L h^−1^ and an ethanol flow rate of 4 mL min^−1^. The optimal sample size of 600 g of pulverized white ray florets was determined in preliminary tests; sample sizes greater than 600 g showed decreased yield (−12.40% yield decrease with 700 g input; −19.65% yield decrease with 800 g input), most likely due to overpacking causing uneven solvent distribution in the extraction tank. The optimal extraction duration time of 2 h was determined in preliminary tests (30 min yields 4.32 ± 0.31 g 100 g^−1^ DW input, 60 min yields 5.86 ± 0.31 g 100 g^−1^ DW input, 90 min yields 9.52 ± 0.68 g 100 g^−1^ DW input, 120 min yields 12.91 ± 0.48 g 100 g^−1^ DW input, and 180 min yields 13.09 ± 0.57 g 100 g^−1^ DW input). The preliminary tests showed that optimal extraction conditions based on yield were using 25 MPa pressure and 55 °C temperature in the extraction tank with no interaction between the two parameters (*p* < 0.05); however, these tests did not take into account the variation of pressure and temperature in the separation tanks. As a result, three different separation temperature (35–55 °C) and separation pressure (5–9 MPa) variations were tested in this study (FAF1 at 45 °C and 7 MPa, FAF2 at 35 °C and 5 MPa, and FAF3 at 55 °C and 9 MPa, respectively). After extraction, the obtained extract was placed in dark glass vials, sealed, and stored at 4 °C to prevent possible degradation. The extraction yield was determined gravimetrically and expressed as g of extract per 100 g of dry white ray florets.

### 3.3. Chemicals and Reagents

The CO_2_ used in SCFE was food grade (purity >99.8%) and purchased from Elmemesser (Riga, Latvia). Ethanol (96%) used in SCFE was purchased from Kalsnavas elevators Ltd. (Jaunkalsnava, Latvia).

LC-MS grade acetonitrile, methanol, and formic acid were purchased from Fisher Scientific (Loughborough, UK), and water for LC-qTOF-MS analysis was purified using a Smart2Pure water purification system (Thermo Scientific, Germany). Gallic acid and AlCl_3_, cyclohexane, Trolox, ferulic acid, rutin, chlorogenic acid were purchased from Acros Organics (Geel, Belgium), Na_2_CO_3_ and NaNO_2_ were obtained from Honeywell (Charlotte, NC, USA), apigenin standard from Rotichrom, Carl Roth GmbH (Karlsruhe, Germany), tannic acid, 2,2-diphenyl-1-picrylhydrazyl (DPPH), phenol, coumarin from Alfa Aesar (Kandel, Germany). Folin–Ciocalteu reagent, H_2_SO_4_, HNO_3_, HCl, D-glucose, NaOH reagents, multi-element standard solution for ICP (33 elements) was purchased from Fisher Scientific (Loughborough, UK). A calibration mix with the major elements Ca, Fe, K, Na, and Mg was purchased from Agilent Technologies (Santa Clara, CA, USA).

Mueller–Hinton broth was purchased from Biolife (Milan, Italy), malt extract from Oxoid (Cheshire, UK), and Wilkins–Chalgren broth from Sigma (St Louis, MO, USA). DMEM medium was purchased from Sigma (Irvine, UK); penicillin–streptomycin as well as calf serum broth were purchased from Sigma (St Louis, MO, USA); and phosphate-buffered saline was purchased from Sigma (Irvine, UK). Neutral Red dye and glacial acetic acid were purchased from Sigma (Irvine, UK).

### 3.4. Quantification of Essential Oils and Waxes

A specific amount of the extract was placed on Petri plates and placed in a 160 °C oven with air circulation for 6 h. Afterwards, the Petri plates were allowed to cool down in a desiccator and weighed. This procedure was repeated until the mass of the extract was constant.

### 3.5. Determination of the Total Phenolic Content

The amount of total phenolic content (TPC) in the studied extracts was determined using a slightly modified Folin–Ciocalteu method [45]. In total, 25 μL of a known dilution of the extract was mixed with 75 μL H_2_O and 25 μL Folin–Ciocalteu reagent (previously diluted 10-fold with deionized water) and allowed to react for 6 min. Then, 100 μL of a 7% solution of sodium carbonate was added to the well. The plates were shaken for 30 s and allowed to stand for 90 min for color development in a dark place at room temperature. Absorbance was measured at 765 nm by using the Epoch 2 UV/VIS Microplate Spectrophotometer (BioTek, Agilent, Germany). The measurements were compared with a standard curve of prepared gallic acid solutions (0–0.2 mg/mL). The total phenolic content was expressed as gallic acid equivalent (GAE) mg. All measurements were performed in triplicates.

### 3.6. Determination of the Total Flavonoid Content

The determination of total flavonoid content (TFC) was performed according to the AlCl3 colorimetric method with slight modifications [46]. 20 μL of a known dilution of the extract was mixed with 15 μL of a 5% solution of sodium nitrite and allowed to react for 5 min in a dark place. Then, 15 µL of 10% aluminum (III) chloride was added to react for 6 min. 100 µL of 1M sodium hydroxide was added to the well after 6 min. The plates were shaken for 30 s and allowed to stand for 15 min for color development in a dark place at room temperature. The reaction forms orange or pink chromophore complexes. Absorbance was measured at 510 nm by using the Epoch 2 UV/VIS Microplate Spectrophotometer (BioTek, Agilent, Germany). The measurements were compared with a standard curve of prepared apigenin solutions (0.025–0.2 mg/mL). The total flavonoid content was expressed as apigenin equivalent (AE) mg. All measurements were performed in triplicates.

### 3.7. Determination of the Total Tannin Content

The number of total tannins (TTC) in the studied extracts was determined using a modified Folin–Ciocalteu method. In total, 50 μL of a known dilution of the extract was mixed with 50 μL H_2_O and 50 μL Folin–Ciocalteu reagent (previously diluted 1:1 with deionized water) and allowed to react for 5 min. Then, 100 μL of 35% sodium carbonate solution (35g Na_2_CO_3_/100 mL H_2_O) was added to the well. The plates were shaken for 30 s and allowed to stand for 30 min for color development in a dark place at the room temperature. The reaction forms a blue chromophore constituted by a phosphotungstic-phosphomolybdenum complex, where the maximum absorption of the chromophores depends on the alkaline solution and the concentration of tannins. Absorbance was measured at 700 nm by using the Epoch 2 UV/VIS Microplate Spectrophotometer (BioTek, Agilent, Germany). The measurements were compared to a standard curve of prepared tannic acid solutions (0.01–0.15 mg/mL). The total tannin content was expressed as tannic acid equivalent (TAE) mg. All measurements were performed in triplicates.

### 3.8. Determination of the Total Sugar Content

For total sugar estimation, the modified Phenol-Sulfuric acid colorimetric method was used. In total, 50 μL of sample, 150 μL of concentrated sulfuric acid (H_2_SO_4_), and 30 μL of 5% phenol (5 g C_6_H_6_O/100 mL H_2_O) reagent were mixed in well and kept in the oven for 5 min at 90 °C. After heating, plates were cooled, and absorbance was measured at 490 nm by using the Epoch 2 UV/VIS Microplate Spectrophotometer (BioTek, Agilent, Germany). The solution turns a yellow-orange color as a result of the interaction between the carbohydrates and the phenol. The absorbance at 490 nm is proportional to the carbohydrate concentration initially present in the sample. The measurements were compared to a standard curve of prepared glucose solutions (0.25–10 mM). The total sugar content was expressed as glucose equivalent (GLE) mg. All measurements were performed in triplicates.

### 3.9. 2,2-Diphenyl-1-picrylhydrazyl (DPPH) Radical Scavenging Assay

The free radical scavenging method DPPH was used to assess the antioxidant properties of tested extracts [45]. For the assay, 20 μL of a known dilution of the extract was mixed with 180 μL of 150 μM DPPH reagent in the wells. The plate was kept in the dark at room temperature for 60 min. When antioxidant samples are mixed with DPPH reagent solution, the color gradually changes from purple to yellow. Decreases in the absorbance at 517 nm were measured using the Epoch 2 UV/VIS Microplate Spectrophotometer (BioTek, Agilent, Germany). Trolox standard solutions in the concentration range of 0–800 μg/mL were used as a standard. All measurements were performed in triplicates. Antiradical activity (ARA) was expressed as trolox equivalent (TE) mg.

### 3.10. Determination of Major and Minor Elements and Heavy Metals by MP-AES Analysis

Approximately 1 g of extract was transferred to a porcelain crucible. Mineralization was conducted in a muffle furnace. The mineralized sample was then transferred to a 50 mL volumetric flask and diluted with deionized water with 37% HCl and 65% HNO_3_. If the concentration of elements was above the detection maximum, samples were diluted with a 5% HNO_3_ solution. Standards were prepared from stock solutions according to Table 7.

Samples were microwave digested on a Berghof Speedwave XPERT DAK-100X (Berghof Products+ Instruments GmbH, Eningen unter Achalm, Germany). The speedwave XPERT Teflon vessels were used for digestion. The digestion program was used as follows: 1st step till 170 °C at a rate of 2 °C/min, hold for 10 min, pressure 80 bar, power 80%; 2nd step to 210 °C at a rate of 3 °C/min, hold for 10 min, power 90%; 3rd step from 210 to 50 °C at a rate of 1 °C/min, hold for 10 min, power 0%. Samples were dry mineralized in a muffle furnace (Nabertherm GmbH, Lilienthal, Germany) for 2h at 650 °C and analyzed on an Agilent 4210 MP-AES with an SPS 4 autosampler and a N2 generator (Agilent Technologies, Deutschland GmbH, Waldbronn, Germany). The Agilent MP Expert acquisition software was used to obtain and analyze the data. Instrument parameters were set as follows: automatic, system-adjusted nebulizer flow for each element (described in Table 8); a read time of 3 s; 3 replicates; pump speed of 15 rpm; an uptake time of 70 s; a rinse time of 45 s; a stabilization time of 25 s; and a correlation coefficient limit of 0.990. The used wavelengths for detected elements are described in Table 8.

### 3.11. UHPLC-HRMS Analysis

Dilutions with ethanol (96%) were prepared to obtain a concentration of at least 10 mg/mL, the samples were filtered through a 0.45 µm filter and were afterwards injected into the chromatographic system. The obtained extracts were analyzed on an Agilent 1290 Infinity II series HPLC system combined with an Agilent 6530 qTOF MS system (Agilent Technologies, Deutschland GmbH, Waldbronn, Germany). Zorbax Eclipse Plus C18 Rapid Resolution HD (2.1 × 150 mm, 1.8 μm particle size) column was used at a flow rate of 0.3 mL/min. The column oven was set at 50 °C, and the sample injection volume was 1 μL with a 30 s needle wash (using 70% methanol). The mobile phase consisted of a combination of A (0.1% formic acid in water) and B (0.1% formic acid in acetonitrile). The gradient elution program was used as follows: Initial 2% B, 0–2 min 2% B, 2–10 min 40% B, 10–20 min 80% B, 20–27 min 95% B, 27–40 min 95% B, 40–42 min 1% B. UV/Vis spectra were recorded at 280 nm and 330 nm. The adjusted operating parameters of the mass spectrometer were set as follows: fragmentation: 70 V; gas temperature: 325 °C; drying gas flow: 10 L/min; nebulizer: 20 psi; sheath gas temperature: 400 °C; and sheath gas flow: 12 L/min. Electrospray ionization (ESI) was used as a source when operating in positive mode. Mass spectra in the *m*/*z* range 50 to 2000 were obtained. The internal reference masses of 121.050873 *m*/*z* and 922.009798 *m*/*z* (G1969-85001 ESI-TOF Reference Mass Solution Kit, Agilent Technologies and Supelco) were used for all analyses of the samples. The Agilent MassHunter Qualitative Analysis 10.0 data acquisition software was applied to analyze LCMS data. The Agilent MassHunter METLIN Metabolomics Database and LipidMaps Database were used for the identification of isolated compounds.

### 3.12. GC-MS Analysis

Samples were diluted in cyclohexane, and extracts were diluted using the mas by volume (1 mg/mL) method. Samples were mixed and filtered through a 0.45 µm filter before being injected into the chromatographic system. Analyses were performed on an Agilent Technologies 7820A gas chromatograph coupled to Agilent 5977B mass selective detector (MSD) equipment. A non-polar HP-5 capillary column (60 m × 0.25 mm, 0.25 µm film thickness) coated with 5% phenyl and 95% methyl polysiloxane. The carrier gas was helium (He) with a split ratio of 1:100 and a flow rate of 1.5 mL/min. The volume of the injection was 3 μL. The temperature program was started at 70 °C, then increased at a rate of 5 °C/min to 230 °C, after which the temperature was increased to 295 °C at a rate of 7 °C/min. Finally, 295 °C was maintained for 30 min. The injector temperature was 270 °C. The mass spectra were recorded at 70 eV. The mass range was 70–500 *m*/*z*. The ion source temperature was maintained at 230 °C. The components were identified based on their retention indices (determined with reference to homologous series of C5–C24 n-alkanes) by comparison of their mass spectra with those stored in the NIST (National Institute of Standards and Technology) MS Search 2.2 library. The Agilent MassHunter Qualitative Analysis 10.0 data acquisition software was applied to analyze GC-MS data. The content of separated compounds was calculated in peak areas using the normalization method without correction factors.

### 3.13. Antimicrobial Activity

Mueller–Hinton broth was used for susceptibility testing by a two-fold serial broth microdilution assay of *Staphylococcus aureus * ATCC 6538P*, Pseudomonas aeruginosa* ATCC 9027, and *Escherichia coli* ATCC 25922. A malt extract broth was used for the testing of *Candida albicans* ATCC 10261.

The inoculum of microorganisms was prepared in sterile water with a density of 0.08–0.10 at A625 and diluted 100-fold in an appropriate broth. Then, 96-well plates were incubated at 37 °C for 24 h. The MIC was determined as the lowest concentration of the studied material, which showed no visible growth of microorganisms. From wells where growth was not detected, 4 μL of media was seeded on appropriate solidified media for MBC/MFC determination.

### 3.14. Cytotoxicity

BALB/3T3 cells (ATCC) (passages 14–15) were seeded in 96-well microplates at a density of 4 × 103 cells/well. Cells were propagated in 100 μL S10 medium (DMEM medium supplemented with 1% penicillin (100 U/mL)–streptomycin (100 μg/mL) (P/S), and 10% calf serum (CS) and incubated overnight at 37 °C, 5% CO_2_. Cells were rinsed with phosphate-buffered saline (PBS), and 100 μL of S5 medium (DMEM medium supplemented with 1% P/S and 5% CS) and extract mix were added to the cells. Additionally, wells with vehicle (appropriate solvent), S5 medium, and sodium dodecyl sulfate (SDS) in S5 medium controls were prepared. After 48 h of incubation at 37 °C with 5% CO_2_, the cells were rinsed with PBS, and 250 μL of 25 μg/mL Neutral Red dye solution in S5 medium was added to all wells. The plate was incubated for 3h at 37 °C, 5% CO_2_, the cells were rinsed with PBS, and 100 μL of NR desorb solution (50% ethanol, 1% glacial acetic acid, 49% water) was added to all wells. The plate was covered and placed in a microplate shaker for 20–45 min, then removed for 5–10min before absorption at 540 nm was measured using a Tecan M200 Infinite Pro microplate reader (Tecan, Switzerland). Cell viability was calculated as a percent of the media control value. The IC50 values were also calculated.

### 3.15. Phototoxicity

The in vitro phototoxicology protocol was a modification of the procedure described in OECD Test Guideline 432 (TG 432). Balb/c 3T3 fibroblast cells were incubated with extracts in 96-well plates for 1 h and thereafter exposed to UVA light (5 J/cm^2^) using UVACUBE 400 (Honle UV Technology, Gilching, Germany). In parallel, cells were exposed to the extracts in the dark and evaluated in parallel. Neutral red dye uptake (NRU) was determined 24 h later, as described in Section 3.14 “Cytotoxicity”.

### 3.16. Statistical Analysis

The chemical data obtained by phytochemical screening using high-throughput 96-well plate methods (Table 2) as well as the composition of volatile phytochemical classes depicted in Figure 4 were expressed as the mean ± standard error of the mean (SEM) of three independent experiments and were analyzed using the computer software GraphPad Prism 8. Statistical analysis of cytotoxicity data was performed using GraphPad Prism 9 software (GraphPad Software, San Diego, California, US). Statistical analysis was performed using Student’s *t*-test (two tailed distribution, two sample equal variances). *p* < 0.05 was considered statistically significant. A heatmap and correlogram with scaled data were created in the R package pheatmap in R software version 4.1.4.(R Foundation Statutes: Vienna, Austria) [47,48,49].

## 4. Conclusions

Management of herbal waste byproducts resulting from various medicinal plant industrial cycles is poorly defined. It is generally recognized that the medicinal plant’s aerial parts have the capacity to serve as natural antioxidants, suggesting that by-products from the industrial processing of medicinal plants may include bioactive compounds that may be used for a variety of applications. This study showed that supercritical fluid extraction (SFE) with CO_2_ as the solvent and ethanol as the co-solvent is a fast, easy-to-automate, and selective way to valorize *Matricaria recutita* white ray florets. Differences in phytochemical content and biological activities of *M. recutita* white floret supercritical fluid extracts subjected to different extraction parameters suggest that their ultimate application potential is strongly dependent on the optimization technique used. Overall results show that *Matricaria recutita* processing by-product white ray floret CO_2_ extracts are rich in biologically active compounds and have cytotoxic and proliferation-reducing activity in immortalized cell lines, as well as antimicrobial activity. This study suggests that white floret extracts of *M. recutita* represent a promising product to produce in specific test systems to fully elucidate the mechanisms of action and possibilities for future applications, such as potential use in cosmetic formulations.

## Figures and Tables

**Figure 1 plants-12-00396-f001:**
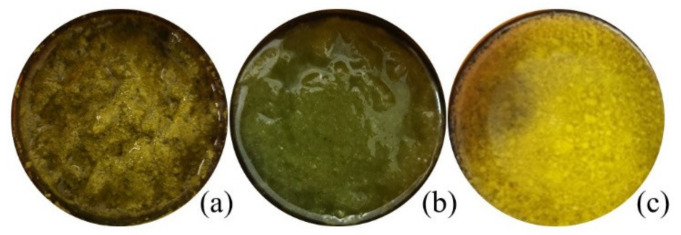
Macroscopic appearance of *Matricaria recucita* white ray floret supercritical fluid extracts under different combinations of temperature and pressure: (**a**) sample FAF1, (**b**) sample FAF2, (**c**) sample FAF3.

**Figure 2 plants-12-00396-f002:**
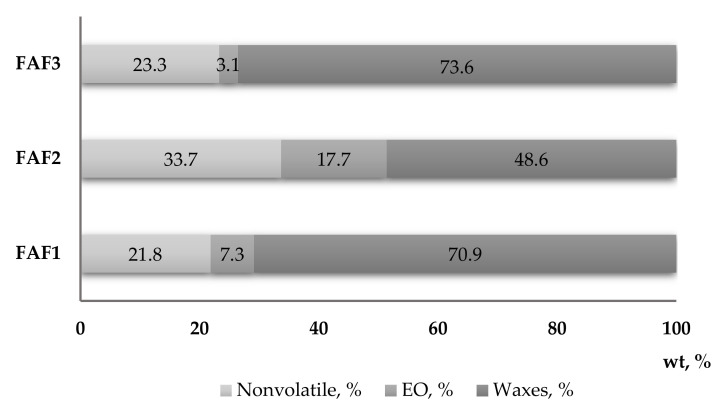
Mass fraction content (wt, %) of essential oils, waxes, and nonvolatile compounds in *Matricaria recutita* white ray floret supercritical fluid extracts.

**Figure 3 plants-12-00396-f003:**
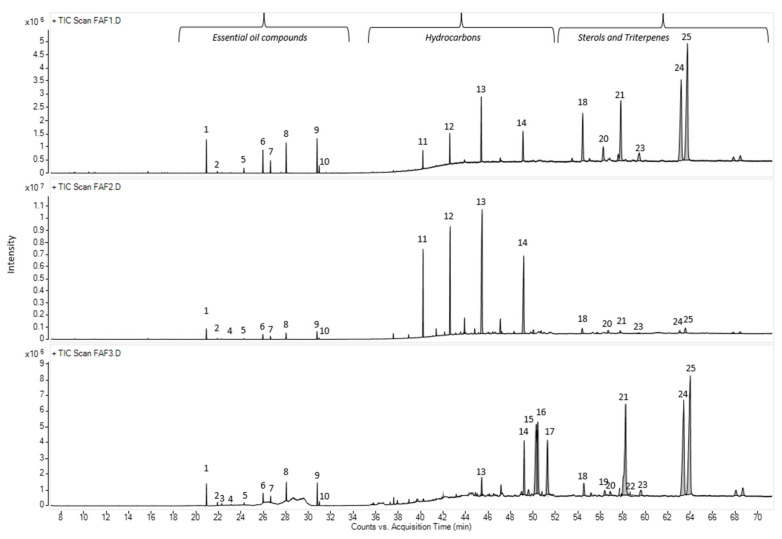
Total ion current chromatogram (TIC) of main volatile compounds in *Matricaria recutita* white ray floret supercritical fluid extracts. The numbers above the peaks represent the main identified compounds in Table 4.

**Figure 4 plants-12-00396-f004:**
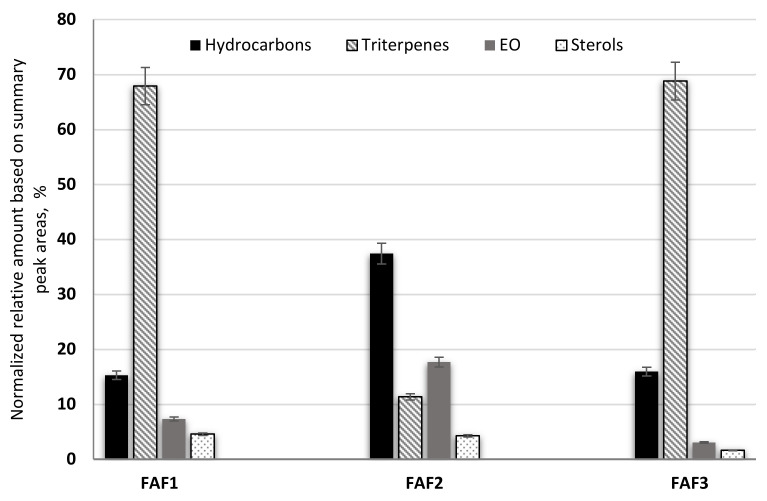
The composition of volatile phytochemical classes in supercritical fluid extracts of *Matricaria recutita* white ray florets in cyclohexane.

**Figure 5 plants-12-00396-f005:**
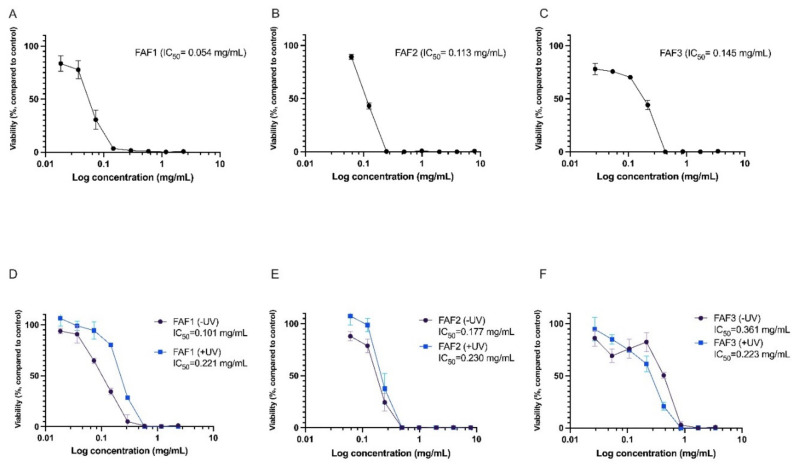
Cytotoxicity (**A**–**C**) and phototoxicity (**D**–**F**) assessment of *Matricaria recutita* white ray floret supercritical fluid extracts. -UV cells incubated with extracts for 1h without UV irradiation, +UV cells preincubated for 1h with extracts and irradiated with 5 J/cm^2^ UVA; n = 3.

**Figure 6 plants-12-00396-f006:**
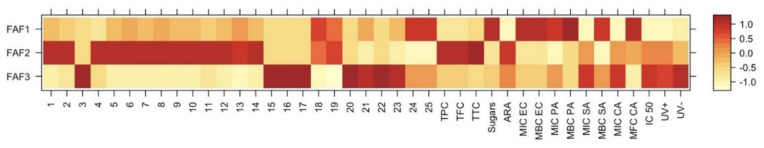
Variation in chemical composition and biological activity of *Matricaria recutita* white ray floret supercritical fluid extracts. The legend denotes scaled values of the volatile chemical constituents (No 1-25, according to Table 4), total phenolics (TPC), flavonoids (TFC), tannins (TTC), sugars (according to Table 2), antiradical (ARA) and antimicrobial activity (according to Table 6), cytotoxicity, and phototoxicity (according to Figure 5).

**Figure 7 plants-12-00396-f007:**
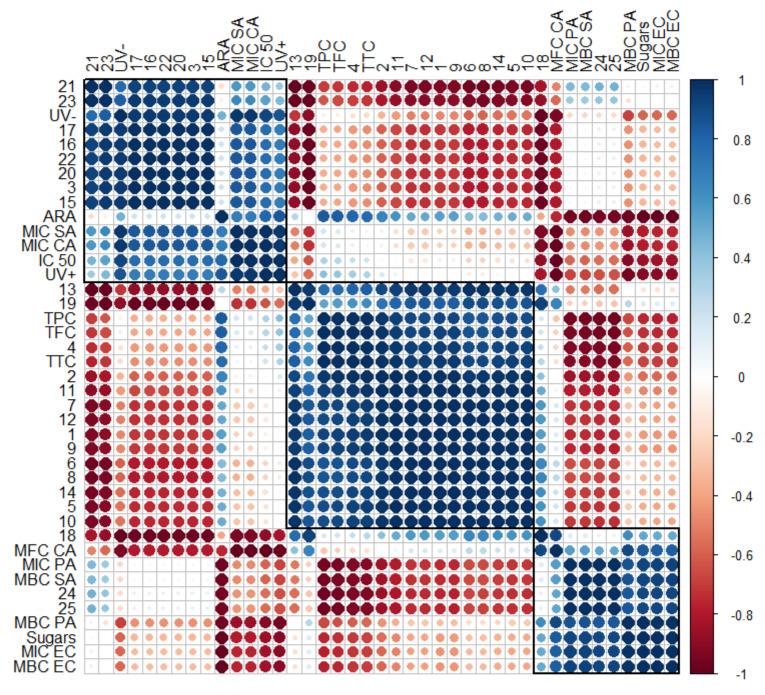
Correlogram depicting the relationship between volatile compounds (No. 1–25, according to Table 4), phytochemical classes (total phenolics (TPC), flavonoids (TFC), tannins (TTC), and sugars, according to Table 2, and their antiradical (ARA) and antimicrobial activity (Table 6), as well as cytotoxicity and phototoxicity (Figure 4). Correlations with *p* > 0.05 are considered insignificant and are denoted by a white space with a blank. The color and size of the squares are proportional to the correlation coefficients, which are color-coded from deep red (−1) to deep blue (1).

**Figure 8 plants-12-00396-f008:**
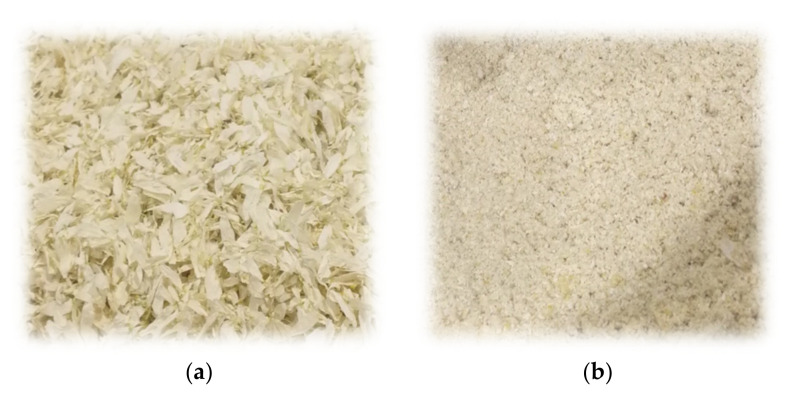
*Matricaria recutita* white ray florets before (**a**) and after (**b**) milling used for supercritical fluid extraction.

**Figure 9 plants-12-00396-f009:**
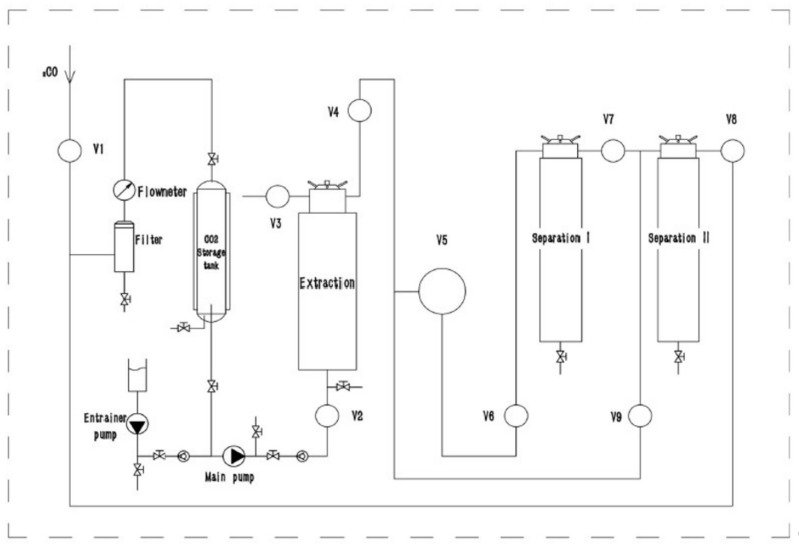
Process diagram of pilot-scale supercritical fluid extractor CAREDDI SCFE-5L.

**Table 1 plants-12-00396-t001:** Extraction yields of *Matricaria recutita* white ray floret supercritical fluid extracts.

Sample	FAF1	FAF2	FAF3
Characterization	Yellow-green, slightly viscous	Bright green, slightly gelatinous	Bright yellow, runny
Yield, g 100 g^−1^ DW input	13.51 ± 0.46	9.76 ± 0.76	18.21 ± 0.15

**Table 2 plants-12-00396-t002:** Content of total phenolics, flavonoids, tannins, sugars, and DPPH free radical scavenging activity of *Matricaria recutita* white ray floret supercritical fluid extracts in ethanol.

Sample	TPC ^a^, GAE mg/mL	TFC ^b^, APE mg/mL	TTC ^c^, TAE mg/mL	Sugars ^d^, GLE mg/mL	ARA ^e^, TE mg/mL	DPPH ^f^ Quenched, %
FAF1	98.1 ± 3.5	45.6 ± 2.7	46.4 ± 1.9	104.0 ± 3.6	3.87 ± 0.09	5.4%
FAF2	342.8 ± 17.3	201.8 ± 11.3	128.8 ± 12.4	53.0 ± 2.2	138.26 ± 0.07	66.5%
FAF3	138.1 ± 3.7	66.0 ± 5.1	49.8 ± 2.7	62.9 ± 3.1	95.06 ± 0.01	16.7%

^a^ Total phenolic content is expressed as the gallic acid equivalents per milliliter of extract (mg GAE/mL). ^b^ Total flavonoid content is expressed as the apigenin equivalents per milliliter of extract (mg APE/mL). ^c^ Total tannin content is expressed as the tannic acid equivalents per milliliter of extract (mg TAE/mL). ^d^ Total sugar content is expressed as the glucose equivalents per milliliter of extract (mg GLE/mL). ^e^ ARA radical scavenging activity is expressed as the trolox equivalents per milliliter of extract (mg TE/mL). ^f^ DPPH radical scavenging activity of 1% extracts (on a dry basis) expressed in %.

**Table 3 plants-12-00396-t003:** Separated phytocomponents from the ethanol extracts of *Matricaria recutita* white ray floret supercritical fluid extracts.

No	Proposed Compounds	Tentative Molecular Formula	Theoretical (*m*/*z*)	Observed (*m*/*z*)	Mass Error (ppm)	Sample FAF
1	Benzoic acid	C_7_H_6_O_2_	123.0441	123.0439	−1.63	1,2
2	Leucine/Isoleucine	C_6_H_13_NO_2_	132.1019	132.1018	−0.76	1,2
3	Furane	C_6_H_8_O_3_	129.0546	129.0549	2.32	1,2
4	Norfuraneol	C_5_H_6_O_3_	115.0390	115.0394	3.48	1,2
5	Phenylalanine	C_9_H_11_NO_2_	166.0863	166.0860	−1.81	1,2
6	N4-Acetylaminobutanal	C_6_H_11_NO_2_	130.0863	130.0862	−0.77	1,2
7	Pyrogallol	C_6_H_6_O_3_	127.0390	127.0391	0.79	1,2
8	Tyrosine	C_9_H_11_NO_3_	182.0812	182.0812	0.00	1,2
9	3-Hydroxy-4,5-dimethyl-2(5H)-furanone	C_6_H_8_O_3_	129.0546	129.0544	−1.55	1,2
10	Trans-3-Hexenyl acetate	C_8_H_14_O_2_	143.1067	143.1068	0.70	1,2
11	Caprolactam	C_6_H_11_NO	114.0913	114.0915	1.75	1,2
12	Ethyl maltol	C_7_H_8_O_3_	141.0546	141.0544	−1.42	1,2
14	Cynaustine	C_15_H_26_C_l_NO_4_	320.1623	320.1622	−0.31	1,2
15	N-(1-deoxy-D-fructos-1-yl)-L-Histidine	C_12_H_19_N_3_O_7_	318.1296	318.1298	0.63	1,2
16	Tryptophyl-Proline	C_16_H_19_N_3_O_3_	302.1499	302.1500	0.33	1,2
18	1-Acetoxypinoresinol	C_22_H_24_O_8_	417.1544	417.1542	−0.48	1,2,3
19	Niacin	C_6_H_5_NO_2_	124.0393	124.0393	0.00	3
20	Methyl hexadecanoate	C_17_H_34_O_2_	271.2632	271.2603	−0.74	3
19	3-(1,1-dimethylallyl)herniarin	C_15_H_16_O_3_	245.1172	245.1173	0.41	1,2,3
20	7-O-Acetyllycopsamine-N-oxide A	C_17_H_27_NO_7_	358.1860	358.1862	0.56	1,2,3
21	Austricin	C_15_H_18_O_4_	149.0597	149.0599	1.34	3
22	Corchoinoside B	C_19_H_28_O_9_	263.1278	263.1277	−0.38	3
23	Cinnamic acid	C_9_H_8_O_2_	401.1806	401.1804	−0.50	1,2,3
24	Prolyl-Tryptophan	C_16_H_19_N_3_O_3_	255.1955	255.1958	1.18	1,2
25	4,4’’-bis(N-feruloyl)serotonin	C_40_H_38_N_4_O_8_	302.1499	302.1502	0.99	1,2
26	Stilbene	C_14_H_12_	703.2762	703.2766	0.57	1,2,3
27	Umbelliferone	C_9_H_6_O_3_	181.1012	181.1009	−1.66	1,2,3
28	Chrysanthetriol	C_15_H_26_O_3_	163.0390	163.0398	4.91	1,2,3
29	Dihydrocaffeic acid	C_9_H_10_O_4_	183.0652	183.0650	−1.09	1,2
30	Loquatoside	C_20_H_22_O_11_	439.1235	439.1230	−1.14	1,2
31	p-Coumaric acid	C_9_H_8_O_3_	165.0546	165.0542	−2.42	1,2,3
32	Isoferulic acid	C_10_H_10_O_4_	195.0652	195.0650	−1.03	1,2,3
33	Epicatechin 3-gallate	C_22_H_18_O_10_	443.0973	443.0970	−0.68	2,3
34	Chrysoeriol 7-rutinoside	C_28_H_32_O_15_	609.1814	609.1819	0.82	3
35	Hesperidin	C_28_H_34_O_15_	611.1970	611.1990	3.27	3
36	Caffeic acid derivative	C_21_H_30_O_13_	491.1759	491.1763	0.81	3
37	Coumaric acid pentoside hexoside	C_20_H_26_O_12_	459.1497	459.1499	0.44	3
38	Catechin derivative	C_15_H_14_O_6_	291.0863	291.0867	1.37	3
39	4-Hydroxy-3,5,4’-trimethoxystilbene	C_17_H_18_O_4_	287.1278	287.1281	1.04	1,2
40	Myricetin 3,3’-digalactoside	C_27_H_30_O_18_	643.1505	643.1511	0.93	1,2
41	Apigenin 6-C-glucoside	C_21_H_20_O_10_	433.1129	433.1135	1.39	1,2
42	Isomucronulatol	C_17_H_18_O_5_	303.1227	303.1234	2.31	1,2
43	Epigallocatechin 3-cinnamate	C_24_H_20_O_8_	437.1231	437.1238	1.60	1,2
44	(-)-Epigallocatechin 7-O-glucuronide	C_21_H_22_O_13_	483.1133	483.1128	−1.03	2
45	Glyphoside	C_23_H_22_O_13_	507.1133	507.1120	−2.56	2
46	Apigenin 7-O-malonylglucoside	C_24_H_22_O_13_	519.1133	519.1138	0.96	1,2,3
47	Myricetin 3-glucoside	C_21_H_20_O_13_	481.0977	481.0982	1.04	2
48	Oleacein	C_17_H_20_O_6_	321.1333	321.1339	1.87	2,3
49	Herniarin	C_10_H_8_O_3_	177.0546	177.0551	2.82	1,2,3
50	B3 (-)-gallocatechin-(4alpha-8)-(+)-catechin	C_30_H_26_O_13_	595.1446	595.1452	1.01	1, 3
51	9-Hexadecenoic acid	C_16_H_30_O_2_	255.2319	255.2325	2.35	1,2
52	Isolariciresinol 3-glucoside	C_26_H_34_O_11_	523.2174	523.2179	0.96	1,2,3
53	(2E,6E)-1-Hydroxy-2,6,10-farnesatrien-9-one	C_15_H_24_O_2_	237.1849	237.1843	−2.53	1,2
54	Muscomin	C_18_H_18_O_7_	347.1125	347.1120	−1.44	2
55	2,3-Dihydroabscisic alcohol	C_15_H_24_O_3_	253.1798	253.1794	−1.58	1,2
56	Phenylethylbenzoate	C_15_H_14_O_2_	227.1067	227.1060	−3.08	1,2
57	Giberellin A	C_19_H_24_O_5_	333.1697	333.1695	−0.60	1,2,3
58	2alpha-Hydroxyalantolactone	C_15_H_20_O_3_	249.1485	249.1480	−2.01	1,2
59	Methyl hexadecanoate	C_17_H_34_O_2_	271.2632	271.2627	−1.84	1,2
60	Arctiopicrin	C_19_H_26_O_6_	351.1802	351.1798	−1.14	1,2,3
61	Chrysosplenol	C_18_H_16_O_8_	361.0918	361.0922	1.11	3
62	Isosyringinoside	C_23_H_34_O_14_	535.2021	535.2027	1.12	3
63	Carnosic acid	C_20_H_28_O_4_	333.2060	333.2052	−2.40	3
64	Matricarin	C_17_H_20_O_5_	305.1384	305.1389	1.64	1,2,3
65	β -D-Xylopyranosyl-(1->4)-a-L-rhamnopyranosyl-(1->2)-D-fucose	C_17_H_30_O_13_	443.1759	443.1762	0.68	1,2
66	Cynaroside A	C_21_H_32_O_10_	445.2068	445.2069	0.22	1,2,3
67	9-Oxohexadecanoic acid	C_16_H_30_O_3_	271.2268	271.2264	−1.47	1,2
68	Retinol	C_20_H_30_O	287.2369	287.2372	1.04	3
69	Unidentified	C_17_H_24_O_3_	277.1798	277.1799	0.36	1,2
70	1-octen-3-ol-3-O-β-D-xylopyranosyl(1->6)- β -D-glucopyranoside	C_19_H_34_O_10_	423.2225	423.2230	1.18	1,2
71	Pinoresinol	C_20_H_22_O_7_	375.1438	375.1443	1.33	1,2,3
72	(E)-En-yn-dicycloether	C_13_H_12_O_2_	201.0910	201.0911	0.50	1,2
73	Cubebininolide	C_24_H_30_O_8_	447.2013	447.2014	0.22	1,2
74	Corchoionoside B	C_19_H_28_O_9_	401.1806	401.1807	0.25	1,2
75	9-Oxooctadecanoic acid	C_18_H_34_O_3_	299.2581	299.2583	0.67	1,2,3
76	Spathulenol	C_15_H_24_O	221.1900	221.1902	0.90	1,3
77	Unidentified	C_17_H_26_O_4_	295.1904	295.1902	−0.68	1
78	Methyl dihydrojasmonate	C_13_H_22_O_3_	227.1642	227.1640	−0.88	1,2
79	Isospathulenol	C_15_H_24_O	815.3332	815.3330	−0.25	1,2,3
80	Tricrocin	C_38_H_54_O_19_	221.1900	221.1901	0.45	3
81	Camelledionol	C_29_H_44_O_3_	441.3363	441.3365	0.45	3
82	Matricin	C_17_H_22_O_5_	307.1540	307.1544	1.30	3
83	13’-Hydroxy-alpha-tocotrienol	C_29_H_44_O_3_	441.3363	441.3363	0.00	3
84	Oleanolic acid	C_30_H_48_O_3_	457.3676	457.3679	0.66	3
85	Stigmasterol	C_29_H_48_O	413.3778	413.3781	0.73	3
86	3-Hydroxy-1,10-bisaboladien-9-one	C_15_H_24_O_2_	237.1849	237.1842	−2.95	1,2
87	Farnesene	C_15_H_24_	205.1951	205.1955	1.95	1,2
88	Acetyl tributyl citrate	C_20_H_34_O_8_	403.2326	403.2327	0.25	1,2
89	Glycerophospate	C_24_H_41_O_7_P	473.2663	473.2660	−0.63	1,2,3
90	Uzarigenin 3-[xylosyl-(1->2)-rhamnoside]	C_34_H_52_O_12_	653.3532	653.3530	−0.31	2
91	β-tocotrienol	C_28_H_42_O_2_	411.3258	411.3260	0.49	1,2
92	ɣ-Tocotrienol	C_28_H_42_O_2_	411.3258	411.3259	0.24	1,2
93	Camellioside D	C_54_H_88_O_24_	1121.5738	1121.5725	−1.16	2
94	17-β-Hydroxy-2alpha-(methoxymethyl)17-methyl-5lapha-androstan-3-one	C_22_H_36_O_3_	349.2737	349.2730	−2.00	1,2
95	Zeaxanthin	C_40_H_52_	533.4142	533.4144	0.37	1,2
96	Lutein	C_40_H_52_	533.4142	533.4148	1.12	1,2

**Table 4 plants-12-00396-t004:** Chemical composition (%) of the cyclohexane extracts from *Matricaria recutita* white ray floret supercritical fluid extracts.

No	Compound ^b^	RI ^a^	FAF1	FAF2	FAF3	Formula ^b^	Class
1	(E)-β-Farnesene	1444	2.10	5.90	0.67	C_15_H_24_	Sesquiterpenoids
2	Germacrene D	1481	0.19	0.75	0.11	C_15_H_24_	Sesquiterpenoids
3	β-Selinene	1486	n.d.	n.d.	0.06	C_15_H_24_	Sesquiterpenoids
4	Bicyclogermacrene	1495	n.d.	0.37	0.03	C_15_H_24_	Sesquiterpenoids
5	Spathulenol	1576	0.21	0.48	0.08	C_15_H_24_O	Sesquiterpenoids
6	α-Bisabolol oxide B	1655	1.10	2.28	0.34	C_15_H_26_O_2_	Tetrahydrofurans
7	α-Bisabolone oxide A	1679	0.65	1.65	0.22	C_15_H_26_O_2_	Tetrahydrofurans
8	α-Bisabolol oxide A	1744	1.38	2.53	0.65	C_15_H_26_O_2_	Tetrahydrofurans
9	cis-ene-yne-Dicycloether	1849	1.39	3.11	0.75	C_13_H_12_O_2_	Spiroethers
10	(E)-Tonghaosu	1902	0.29	0.60	0.14	C_13_H_12_O_2_	Spiroethers
11	Tetracosane	2400	2.27	10.54	n.d.	C_24_H_50_	Hydrocarbons
12	Hexacosane	2600	2.90	9.60	n.d.	C_26_H_54_	Hydrocarbons
13	Heptacosane	2700	6.52	12.08	0.21	C_27_H_56_	Hydrocarbons
14	Octacosane	2800	3.59	5.20	2.73	C_28_H_58_	Hydrocarbons
15	Nonacosane	2900	n.d.	n.d.	3.11	C_29_H_60_	Hydrocarbons
16	Tritriacontane	3300	n.d.	n.d.	6.23	C_33_H_68_	Hydrocarbons
17	Tetratriacontane	3400	n.d.	n.d.	3.69	C_34_H_70_	Hydrocarbons
18	Stigmasterol	3170	3.35	2.87	1.12	C_29_H_48_O	Sterols
19	Sitosterol	3173	1.22	1.39	0.51	C_29_H_50_O	Sterols
20	Taraxerol	2834	n.d.	n.d.	3.37	C_30_H_50_O	Triterpenes
21	Lupeol	3270	10.58	2.56	23.26	C_30_H_50_O	Triterpenes
22	Ψ-Taraxasterol	3290	n.d.	n.d.	0.7	C_30_H_50_O	Triterpenes
23	Taraxasterol	3293	2.31	0.69	5.9	C_30_H_50_O	Triterpenes
24	α-Amyrin	3328	24.04	2.97	15.32	C_30_H_50_O	Triterpenes
25	β-Amyrin	3337	30.97	5.13	20.24	C_30_H_50_O	Triterpenes
	Other compounds		4.94	29.3	10.56		

^a^ Retention indexes (RI) determined on the HP-5MS capillary column. ^b^ Based on NIST (National Institute of Standards and Technology) MS Search 2.2 library. n.d.—not detected.

**Table 5 plants-12-00396-t005:** Detection of major, minor elements, and heavy metals in *Matricaria recutita* white ray floret supercritical fluid extracts.

Elements, mg/kg	FAF1	FAF2	FAF3	Mean	Min	Max	CV, %
Ca	510.0	503.1	507.2	506.8	503.1	510	0.7
Fe	1170.0	1189.3	1182.2	1180.5	1170	1189.3	0.8
K	0.17	0.18	0.18	0.2	0.17	0.18	3.3
Mg	50.0	52.2	55.9	52.7	50	55.9	5.7
Na	110.0	104.3	109.0	107.8	104.3	110	2.8
Cu	3.8	3.6	3.5	3.6	3.5	3.8	4.2
Ni	10.7	10.5	10.2	10.5	10.2	10.7	2.4
Mn	4.1	4.3	4.1	4.2	4.1	4.3	2.8
Zn	1.8	1.9	1.7	1.8	1.7	1.9	5.6

**Table 6 plants-12-00396-t006:** Minimum inhibitory concentrations (MIC), minimum bactericidal (MBC), and minimum fungicidal (MFC) concentrations, mg/mL of *Matricaria recutita* white ray floret supercritical fluid extracts.

Sample	*E. coli*		*P. aeruginosa*		*S. aureus*		*C. albicans*	
	MIC	MBC	MIC	MBC	MIC	MBC	MIC	MFC
FAF1	2.94	2.94	0.74	1.47	0.37	0.74	0.37	0.74
FAF2	4.95	4.95	2.48	4.95	0.31	2.48	0.31	1.23
FAF3	4.30	4.30	1.08	4.30	0.27	1.08	0.27	2.15

**Table 7 plants-12-00396-t007:** Concentration values of standard solutions prepared for MP-AES analysis.

Element	Flask Volume for Each Concentration, mL	Stock Solution, mg/L	Calibration Range
Ca, Fe, K, Mg, Na mix solution	50 mL, 5% HNO_3_	500 mg/L	0.25; 0.50; 1.0; 2.5; 4.0; 5.0; 7.5 mg/L (at least R^2^ = 0.990)
Cd, Co, As, Cr, Cu, Mn, Mo, Ni, Pb, Zn mix solution	50 mL, 5% HNO_3_	5 mg/L	0.1; 0.25; 0.5; 1.0; 2.5; 5 mg/L (at least R^2^ = 0.990)

**Table 8 plants-12-00396-t008:** Detection conditions for micro- and macro-elements by atomic emission spectrometer.

Element	Wavelength, nm	Nebulizer Flow, L/min	Element	Wavelength, nm	Nebulizer Flow, L/min
Na	588.995	0.95	Cr	425.433	0.90
Mg	285.213	0.90	Cu	324.754	0.70
K	766.491	0.75	Mn	403.076	0.90
Ca	393.366	0.60	Mo	379.825	0.85
Fe	371.993	0.65	Ni	361.939	0.70
Cd	226.502	0.50	Pb	283.305	0.75
Co	340.512	0.75	Zn	213.857	0.45
As	197.198				

## Data Availability

The data presented in this study are available on request from the corresponding authors.

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
