# Peer review of "Valorization of Bioactive Compounds from By-Products of Matricaria recutita White Ray Florets"

_plants, 2023, doi:10.3390/plants12020396_

Round 1
Reviewer 1 Report
The manuscript: “Valorization of Bioactive Compounds from By-products of Matricaria Recutita White Ray Florets” by Nakurte I et al, is an interesting paper focused on a suitable green strategy to identify the chemical composition of the extracts from Matricaria Recutita White Ray Florets using supercritical fluid extraction (SFE) with CO2. The results showed that Chamomile processing by-product white ray florets CO2 extracts are rich in biologically active compounds with both cytotoxic and proliferation-reducing activity in immortalized cell lines, and antimicrobial activity.
This study can open a new scenario for identifying bio-based compounds with efficient eco-friendly methods.
The paper is well-written, and the text is clear and easy to read. In my opinion, the manuscript is suitable for publication in Plants
Author Response
Dear Reviewer!
Thank you for the time you have invested in reviewing our manuscript. Thank you very much for such a positive rating. I want to inform you that following the recommendations of the other three reviewers, we have made several improvements to our manuscript. The improved manuscript can be found in the attached file.
Wishing you all the best,
Ilva Nakurte, Dr.chem.
Institute for Environmental Solutions
Leading researcher

Reviewer 2 Report
The Paper Valorization of Bioactive Compounds from By-products of 2 Matricaria Recutita White Ray Florets shows a very precise breakdown of the SPE extract from the white ray florets of Matricaria Recutita, as well as promising results regarding cytotoxicity and phototoxicity of the same. However, precise details on the statistical analysis applied are missing. Furthermore, a conclusion in the form of a separate chapter would be desirable (unless otherwise requested by MDPI), partly this is already included in chapter 2.7. Overall and especially in the third chapter I would like to see more figures e.g. in the third chapter of the applied gradients, obtained chromatograms, flow sheets of the used apparatus, …. The text shows also major spelling mistakes and I would highly recommend, that a native speaker should re-read the text.
Please find further comments on the paper in the attached word file.
Wish you all a happy new year!

Author Response
Response to Reviewer 2 Comments
Dear Reviewer,
thank you kindly for your thorough approach to the textual corrections as well as your thoughtful comments, recommendations, and questions! We have attached a pdf file of the improved manuscript, showing all the changes we made. Please find our answers and explanations below.
Comments: However, precise details on the statistical analysis applied are missing.
Response: According to your suggestion, we have improved Section 3.17 (page 21).
Comments: Furthermore, a conclusion in the form of a separate chapter would be desirable (unless otherwise requested by MDPI), partly this is already included in chapter 2.7.
Response: According to your suggestion, we made separate Section 4 with conclusions (page 21).
Comments: Overall and especially in the third chapter I would like to see more figures e.g. in the third chapter of the applied gradients, obtained chromatograms, flow sheets of the used apparatus, ….
Response: According to your suggestion, we improve Section 3 with a process diagram of a pilot-scale supercritical fluid extractor, the CAREDDI SCFE-5L (Figure 9 on page 16), as well as we add
GC-MS chromatograms (Figure 3 on page 8) in Section 2.
Comments: The text shows also major spelling mistakes and I would highly recommend, that a native speaker should re-read the text.
Response: According to your recommendation, we made improvements.
Comments:
Line 41 – delete the “ before and after green chemistry. This sounds sarcastic and not emphasized.
Lines 41-43 - both sentences are not connected. It reads more like a listing.
Line 49 – specify which businesses
Line 53 – use challenge instead of difficulty
Paragraph lines 84-92 – Name some new green extraction methods as well as conventional techniques
Lines 93-95 – Specify why SFE is considered green and also name some examples for the application in the food and pharmaceutical industry and environmental engineering.
Paragraph lines 106-110 – this paragraph would suit better before the SFE paragraph in terms of content
Lines 113-114 - Matricaria recutita white ray florets were collected from a dry herb processing facility
Line 129 – you started using SFE as abbreviation for supercritical fluid extraction but here and in the following paragraphs you use SCFE as abbreviation. Please be consistent in your abbreviations! I read the third chapter before the second (because it was more logic to me, I know this order here is preferred by mdpi) I did not check for the spelling mistakes I mentioned in chapter three in chapter two. So please re-read the text and look for them. Especially for the comment I wrote on paragraph lines 581-586.
Lines 262-265 – name some specific examples were the essential oils from camomile are used not only the industries
Lines 476-477 - The sample size 600 g of pulverized white ray florets and an extraction duration of 2 hours was used and kept constant based on preliminary tests (data not shown).
Line 484 – put blanks between numbers and units (this happens also in other parts in the text) A flow-sheet of the SFE-apparatus would be great!
Line 507 – broth
Lines 560-561 – do not separate numbers and units, use a so-called protected blank (STRG + Shift + space key)
Lines 570-571 - DPPH was used to assess the free radical scavenging method was used to assess (antioxidant) property of tested extracts (Herald et al., 2012). This sentence does not make sense, you finished it different than you started.
Paragraph lines 581-586 – you forget to put a, the etc. several times. The text reads more like bullet points
Lines 583-584 - If the concentration of elements for instrument was above detection maximum, … What do you with for instrument?
Lines 605-606 - Dilutions with ethanol (96%) were prepared to obtain a concentration of at least 10 mg/mL, the samples were filtered through 0.45 μm filter and were afterwards injected into the chromatographic system.
Paragraph lines 605-624 – see comment on paragraph lines 581-586, also a diagram of the gradient would be nice
Lines 627-629 – see comment on paragraph 581-586
Paragraph lines 684-686 – This paragraph needs much more information on how the statistical analysis was performed! Which settings were applied in the statistical analysis? Which algorithms were used and why?
Response: We've made all of the following changes that you suggested based on your comments and suggestions (see the revised manuscript).
Comments: Figure 3: Add a grid, y-axis: what type of %? Wt. %, vol. %, …?
Response: Recommended changes have been made to Figure 3, which is now Figure 4 on page 9.
Figure 4: Again on y-axis: what type of percent? Also the x- and y-axis have a highly different amount of markings, I would suggest a partition every 10 % on the y-axis and one of 0.01, 0.02, 0.03, … 0.1, 0.2, 0.3 … 1, 2, 3 … on the x-axis.
Response: Recommended changes have been made to Figure 4, which is now Figure 5 on page 13.
Figure 5 could be larger figure 6 should be larger. Both figures should also have a better resolution.
Response: Recommended changes have been made to Figures 5 and 6, which are now Figure 6 on page 13 and Figure 7 on page 15, respectively.
Best regards,
Ilva Nakurte, Dr.chem.
Institute for Environmental Solutions
Leading researcher

Reviewer 3 Report
This manuscript contains the results of supercritical fluid extraction (SFE) from Matricaria recutita white ray florets and comparison of the SFE yield and chemical composition of the isolates. In this research, they have presented the valorization possibilities of Matricaria recutita white ray florets using supercritical fluid extraction (SFE) with CO2. Experiments were done at temperatures of 35–55 â—¦C and separation pressures of 5–9 MPa to evaluate their impact on the chemical composition and biological activity of the extracts. The total achieved extraction yields varied from 9.76 to 18.21 g 100 g-1 input.
In general, Few comments for further improvement are,
Comments:
1- The abstract should be informative including main achivments…
2- State the advantages and disadvantages of the extraction techniques mentioned in the Introduction. Also, state the reasons why choose the SFE method.
3- The introduction should be rewritten with better references(Aris, et al. 2018,Putra, et al. 2018,Rai, et al. 2016,Kant and Kumar 2022,Amani, et al. 2021,Sodeifian, et al. 2019)
4- The abbreviations should be presented in a Table.
5- Please explain why to consider the separation conditions as effective parameters. “Therefore, three different variations of separation temperature 482 (35Ëš-55ËšC) and separation pressure (5-9 MPa) were tested within this study (FAF1 45 ËšC 483 and 7MPa, FAF2 35 ËšC and 5MPa, FAF3 55 ËšC and 9MPa, respectively).”
6- A comparison of results (yield, operating conditions and composition) with other studies on different types of extraction should be added. This is to give a better appreciation of the work conducted.
7- Please add more detalies about apparatus and device….
References
1. Aris, N.A., Zaini, A.S., Nasir, H.M., Idham, Z., Vellasamy, Y. and Yunus, M.A.C. 2018 Effect of particle size and co-extractant in Momordica charantia extract yield and diffusion coefficient using supercritical CO2. Malaysian Journal of Fundamental and Applied Sciences, 14 (3), 368-373.
2. Putra, N.R., Rizkiyah, D.N., Zaini, A.S., Yunus, M.A.C., Machmudah, S., Idham, Z.b. et al. 2018 Effect of particle size on yield extract and antioxidant activity of peanut skin using modified supercritical carbon dioxide and soxhlet extraction. Journal of Food Processing and Preservation, 42 (8), e13689.
3. Rai, A., Mohanty, B. and Bhargava, R. 2016 Supercritical extraction of sunflower oil: A central composite design for extraction variables. Food chemistry, 192, 647-659.
4. Kant, R. and Kumar, A. 2022 Review on essential oil extraction from aromatic and medicinal plants: Techniques, performance and economic analysis. Sustainable Chemistry and Pharmacy, 30, 100829.
5. Amani, M., Ardestani, N.S. and Honarvar, B. 2021 Experimental Optimization and Modeling of Supercritical Fluid Extraction of Oil from Pinus gerardiana. Chemical Engineering & Technology, 44 (4), 578-588.
6. Sodeifian, G., Ardestani, N.S. and Sajadian, S.A. 2019 Extraction of seed oil from Diospyros lotus optimized using response surface methodology. Journal of forestry research, 30 (2), 709-719.

Author Response
Response to Reviewer 3 Comments
Dear Reviewer,
thank you for your thoughtful comments, recommendations, and questions! We have attached a pdf file of the improved manuscript, showing all the changes we made. Please find our answers and explanations below.
Comments: 1- The abstract should be informative including main achivments…
Response: According to your suggestion, we have improved abstract (on page 1).
Comments: 2- State the advantages and disadvantages of the extraction techniques mentioned in the Introduction. Also, state the reasons why choose the SFE method.
Response: Recommended changes have been made in the introduction on page 2. The specified information is found in paragraph lines 113-152.
Comments: 3- The introduction should be rewritten with better references(Aris, et al. 2018,Putra, et al. 2018,Rai, et al. 2016,Kant and Kumar 2022,Amani, et al. 2021,Sodeifian, et al. 2019). 4- The abbreviations should be presented in a Table.
Response: The MDPI author guidelines require that references in the introduction be in number format.
4- The abbreviations should be presented in a Table.
The MDPI author guidelines require that “ Acronyms/Abbreviations/Initialisms should be defined the first time they appear in each of three sections: the abstract; the main text; the first figure or table. When defined for the first time, the acronym/abbreviation/initialism should be added in parentheses after the written-out form”. We respected these conditions, so we did not dedicate a separate section to abbrevations.
Comments: 5- Please explain why to consider the separation conditions as effective parameters. “Therefore, three different variations of separation temperature 482 (35Ëš-55ËšC) and separation pressure (5-9 MPa) were tested within this study (FAF1 45 ËšC 483 and 7MPa, FAF2 35 ËšC and 5MPa, FAF3 55 ËšC and 9MPa, respectively).”
Response: The paragraph lines 164–169 in the revised manuscript contain the specified information.
“Based on the reduction in pressure and change in temperature, different compounds will precipitate in each stage of separation. The separation may occur either because the supercritical fluid is no longer supercritical or because the solute is no longer soluble in the supercritical fluid. This step may concentrate and isolate compounds from the extract”.
Comments: 6- A comparison of results (yield, operating conditions and composition) with other studies on different types of extraction should be added. This is to give a better appreciation of the work conducted.
Response: The paragraph lines 194–234 in the revised manuscript contain the specified information.
“The highest overall extraction yields as obtained in the current research is higher than those previously reported. Extract yield between 0.23 and 3.64 g/100 g of different processed and unprocessed chamomile flowers has been reported by Molnar and team [13], while Kotnik [17] and Scalia [18] reported yields of 2.50-3.81 g/100 g and 9.2-9.7g/100 g respectively, obtained from ground chamomile flower heads.”
[13] Molnar, M.; Mendešević, N.; Šubarić, D.; Banjari, I.; Jokić, S. Comparison of various techniques for the extraction of umbelliferone and herniarin in Matricaria recutita processing fractions. Chem Cent J. 2017, 5, 11, 78. https://doi.org/10.1186/s13065-017-0308-y
[17] Kotnik, P.; Škerget, M.; Knez, Ž. Supercritical fluid extraction of chamomile flower heads: Comparison with conventional extraction, kinetics and scale-up. J. of Supercritical fluids 2007, 43, 192-198. doi:10.1016/j.supflu.2007.02.005
[18] Scalia, S.; Guiffreda, L.; Pallado, P. Analytical and preparative supercritical fluid extraction of Chamomile flowers and its comparison with conventional methods. Journal of Pharmaceutical and Biomedical analysis 1999, 21, 549-558. doi:10.1016/s0731-7085(99)00152-1
Comments: 7- Please add more detalies about apparatus and device…
Response: According to your suggestion, we improve Section 3 with a process diagram of a pilot-scale supercritical fluid extractor, the CAREDDI SCFE-5L (Figure 9 on page 16).
Best regards,
Ilva Nakurte, Dr.chem.
Institute for Environmental Solutions
Leading researcher

Reviewer 4 Report
I suggest to rewrite the abstract, indicating some further results ; check some words, such as chamomile and the Latin name of the plant. Remember that the name of the species is not capitalized.
The paper is very complex and, in some moments, difficult to follow.
Author Response
Response to Reviewer 4 Comments
Comments: I suggest to rewrite the abstract, indicating some further results ; check some words, such as chamomile and the Latin name of the plant. Remember that the name of the species is not capitalized.
The paper is very complex and, in some moments, difficult to follow.
Dear Reviewer!
Thank you for the time you have invested in reviewing our manuscript. Thank you very much for the positive rating. I want to inform you that following your recommendations and the comments of the other three reviewers, we have made several improvements to our manuscript. The improved manuscript can be found in the attached file.
Response: According to all suggestions, we have improved the abstract (on page 1). Changes have been made to plant names throughout the manuscript. The Latin name of chamomile (Matricaria recutita) is mostly used in our research. Plant species are not capitalized.
Wishing you all the best,
Ilva Nakurte, Dr.chem.
Institute for Environmental Solutions
Leading researcher
